# Control of morphology and formation of highly geometrically confined magnetic skyrmions

Chiming Jin[1,2], Zi-An Li[3,4,5], András Kovács[3], Jan Caron[3], Fengshan Zheng[3], Filipp N. Rybakov[6,7,8], Nikolai S. Kiselev[9], Haifeng Du[1,10], Stefan Blügel[9], Mingliang Tian[1,10], Yuheng Zhang[1,10], Michael Farle[4,11] & Rafal E. Dunin-Borkowski[3]

The ability to controllably manipulate magnetic skyrmions, small magnetic whirls with particle-like properties, in nanostructured elements is a prerequisite for incorporating them into spintronic devices. Here, we use state-of-the-art electron holographic imaging to directly visualize the morphology and nucleation of magnetic skyrmions in a wedge-shaped FeGe nanostripe that has a width in the range of 45–150 nm. We find that geometrically-confined skyrmions are able to adopt a wide range of sizes and ellipticities in a nanostripe that are absent in both thin films and bulk materials and can be created from a helical magnetic state with a distorted edge twist in a simple and efficient manner. We perform a theoretical analysis based on a three-dimensional general model of isotropic chiral magnets to confirm our experimental results. The flexibility and ease of formation of geometrically confined magnetic skyrmions may help to optimize the design of skyrmion-based memory devices.

[1] The Anhui Key Laboratory of Condensed Matter Physics at Extreme Conditions, High Magnetic Field Laboratory, Chinese Academy of Science (CAS), Hefei, Anhui Province 230031, China. [2] Department of Physics, University of Science and Technology of China, Hefei, Anhui Province 230031, China. [3] Ernst Ruska-Centre for Microscopy and Spectroscopy with Electrons and Peter Grünberg Institute, Forschungszentrum Jülich, Jülich 52425, Germany. [4] Faculty of Physics and Center for Nanointegration (CENIDE), University of Duisburg-Essen, Duisburg 48047, Germany. [5] Institute of Physics, Chinese Academy of Sciences, Beijing 100190, China. [6] M.N. Miheev Institute of Metal Physics of Ural Branch of Russian Academy of Sciences, Ekaterinburg 620990, Russia. [7] Ural Federal University, Ekaterinburg 620002, Russia. [8] KTH Royal Institute of Technology, Stockholm SE-10691, Sweden. [9] Peter Grünberg Institute and Institute for Advanced Simulation, Forschungszentrum Jülich and JARA, Jülich 52425, Germany. [10] Collaborative Innovation Center of Advanced Microstructures, Nanjing University, Jiangsu Province 210093, China. [11] Center of Functionalized Magnetic Materials, Immanuel Kant Baltic Federal University, Kaliningrad 236041, Russian Federation. Correspondence and requests for materials should be addressed to Z.-A.L. (email: zali97@iphy.ac.cn) or to N.S.K. (email: n.kiselev@fz-juelich.de) or to H.D. (email: duhf@hmfl.ac.cn).

Magnetic storage in hard disk drive technology is based on the controllable formation of magnetic domains and is approaching its limits[1]. The ability to manipulate domain walls instead of domains provides an alternative method for further extending the storage device roadmap[2,3]. The discovery of topologically stable magnetic skyrmions is of great interest because of their small size (typically below ~100 nm) and their high mobility at low-current densities[4–10]. In particular, skyrmions are promising candidates for applications in novel data storage devices based on the race-track memory concept[11]. The underlying design of such devices relies on the use of skyrmions as data bit carriers, which move along a ferromagnetic nanostripe that takes on the role of a guiding track. It is therefore important to be able to controllably form and manipulate skyrmions in nanostructured elements. Over the past few years, many theoretical proposals have been put forward to host and create individual skyrmions in confined geometries[12–17], including the possibility of locally nucleating an isolated skyrmion by using a spin-polarized current and driving its motion using a current-induced torque[12,13]. On the experimental side, skyrmion chains and skyrmion cluster states have been observed in FeGe nanostripes of fixed width[18] and nanodisks[19], respectively, using Lorentz transmission electron microscopy (TEM). However, the widths of the samples in these studies were all larger than the corresponding single skyrmion size. Moreover, recent work has demonstrated powerful skyrmion-based information storage functionalities[20], such as the ability to generate skyrmion bubbles in geometrically confined CoFeB/Ta films using in-plane currents[21]. Despite these experimental results, the formation and manipulation of skyrmions has never been studied in detail in confined geometries of sufficiently small dimension. The optimization of information storage requires an optimal width for a nanostripe that hosts a chain of skyrmions, as well as precise control of their morphology[22] and formation in such a confined geometry. Therefore, the next important step is to develop approaches that can be used to control the fine structures of individual skyrmions and to nucleate them in a simple manner in a nanostripe, whose size is similar to that of the skyrmions themselves. The present study is also directly connected to the determination of the smallest width of a nanostripe, which is still able to host magnetic skyrmions over a wide range of applied fields and temperatures.

Real-space imaging of skyrmions is essential for addressing these challenges[6,20–24]. However, the study of individual magnetic skyrmions in nanostructured elements is very difficult because of the required spatial resolution and sensitivity and the influence of the edge of a nanostructure on the recorded contrast. The latter issue affects the study of skyrmions in nanostructures that have sizes of below 100 nm using Lorentz TEM because Fresnel fringes, which are formed at the edge of a nanostructure as a result of the need to use an out-of-focus imaging condition, complicate the interpretation of the recorded magnetic signal[18,25]. In contrast, the technique of off-axis electron holography (EH) in the TEM allows an electro-optical phase image of a specimen to be recorded directly, so that unwanted contributions to the signal from the edges of a nanostructure can be subtracted more easily than using other TEM-based techniques. Moreover, the technique has nm spatial resolution, high phase sensitivity and permits reliable quantification of magnetic states in nanostructured elements[26]. Digital acquisition and analysis of electron holograms and image analysis software are then essential, when studying weakly varying phase objects such as skyrmions. The technique has been used to study the three-dimensional structures of skyrmions in $Fe_{0.5}Co_{0.5}Si$ films, for which a low temperature of ~10 K, far below the Curie temperature $T_c \sim 35$ K, was necessary to obtain a high quality signal[27].

Here, we use off-axis EH to study a wedge-shaped nanostripe, whose width reaches only 40–150 nm, at several different temperatures, in order to investigate the fine structure and formation of highly geometrically confined skyrmions. We observe the high flexibilities of individual skyrmions and a unique field-driven helix-to-skyrmion transition directly.

## Results

**Theoretically predicted skyrmion morphology in a nanostripe.** We first discuss the expected skyrmion morphology in a nanostripe. In general, the stability of skyrmions in chiral magnets, such as MnSi, $Fe_xCo_{1-x}Si$, FeGe and other B20 alloys, is governed by the Dzyaloshinskii–Moriya interaction (DMI)[8]. Competition between the DMI and ferromagnetic exchange coupling results in a homochiral spin helix ground state with an equilibrium period $L_D$, which is determined by the ratio of the energy contributions of these two interactions. In a bulk crystal, the spin helix generally evolves into a conical phase and then into a field-saturated ferromagnetic state in the presence of an increasing external magnetic field. Skyrmions appear in the form of a lattice and occupy only a tiny pocket in the magnetic field $H$ and temperature $T$ phase space, as the temperature is slightly below the Curie temperature $T_c$. In a thin crystal of a chiral magnet, the stability of a skyrmion lattice phase will be significantly enhanced[6,25] due to uniaxial anisotropy[28] or spatial confinement[29,30]. In both bulk compounds and thin films, skyrmion crystals have a fixed lattice constant and adopt approximately circular shapes. Both theoretical analysis[31] and experimental observations[6,25] suggest that elliptical distortions of skyrmions in extended systems are associated with a loss of stability, which is referred to as an elliptical or strip-out instability. However, because the twists of spins within a skyrmion give rise to a non-trivial magnetization topology, it should be possible to tune its geometrical morphology without changing its topological class[9]. Recent Lorentz TEM observations of a FeGe nanostripe with a fixed width of 130 nm suggest that elliptical skyrmions can be supported in such a confined geometry[18]. Simulations performed within the framework of a general model for a three-dimensional isotropic chiral magnet[32] in nanostripes further confirm this hypothesis (see the Methods section). In particular, for a long nanostripe of width $W_y$ on the order of the skyrmion size and length $W_x \gg W_y$, the theoretical model predicts a loss of radial symmetry of skyrmions without a loss of their stability (Fig. 1a). The skyrmion morphology in a single chain is predicted to depend on nanostripe width $W_y$ for a certain range of applied magnetic fields. The dimensions of elliptical skyrmion shapes can be described in terms of their semi-axes $a$ and $b$ along and perpendicular to the nanostripe, respectively. For nanostripe widths $W_y$ close to a critical value $W_y^c$, skyrmions exhibit circular shapes ($a = b$), while for $W_y < W_y^c$ and $W_y > W_y^c$ they are predicted to show longitudinal ($a > b$) and transverse ($a < b$) ellipticity, respectively. On further increasing the nanostripe width above a second critical value $W_y^t$, the skyrmions are expected to arrange themselves in the form of a zigzag chain at equilibrium.

**Experimental results.** To test these predictions, we carefully fabricated a wedge-shaped nanostripe from a bulk crystal of FeGe using a lift-out method (Supplementary Fig. 1). Bulk FeGe possesses a high Curie temperature, $T_c \sim 278$ K (ref. 25). The nanostripe had a thickness $L$ of 110 nm, a length $W_x$ of ~2.6 μm and a width $W_y$ that varied linearly from ~10 to ~180 nm (Fig. 1b and Supplementary Fig. 2). This range of widths is

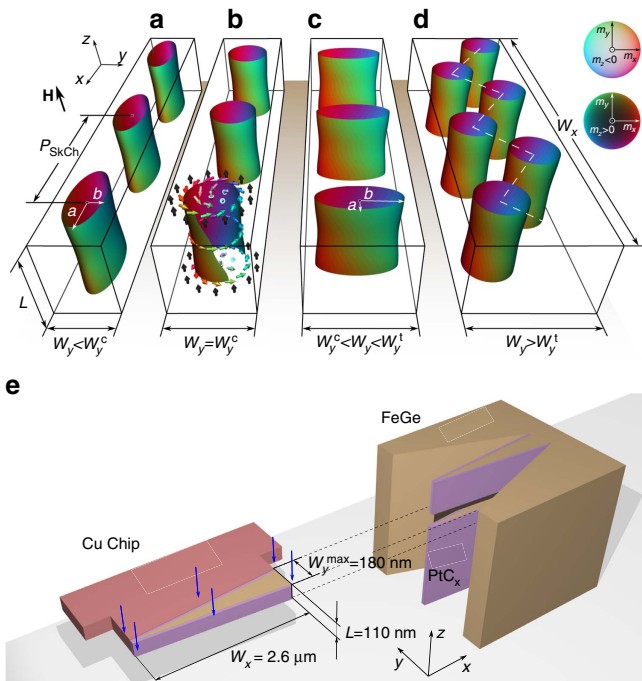

**Figure 1 | Schematic diagrams of geometrical control of skyrmions.**
(**a–d**) Schematic diagrams showing chains of skyrmion tubes in nanostripes of varying width $W_y$. $W_y^c$ is the critical width of a nanostripe, for which a circular skyrmion appears in **b**. Below $W_y^c$, skyrmions show longitudinal ellipticity in **a**, while above $W_y^c$ skyrmions show transverse ellipticity in **c**. $W_y^t$ is a second critical width, above which skyrmions are expected to arrange themselves in the form of a zigzag chain at equilibrium, as shown in **d**. The isosurfaces correspond to a value of zero for the out-of plane magnetization component $m_z$. The colours represent the $m_x$ and $m_y$ components of the magnetization. $a$ and $b$ are the semi-axes of the skyrmions along and perpendicular to the nanostripe, respectively. The spin arrangement in a single skyrmion tube is shown using arrows. The colour wheel in **d** represents the strength and direction of the in-plane magnetization at each point. (**e**) Geometry of the FeGe wedge-shaped sample used to investigate geometrically-confined skyrmions using off-axis EH and Lorentz TEM. The specimen was fabricated using a lift-out method and coated with amorphous PtC$_x$, as described in Supplementary Fig. 1. In the present experiments, both the applied magnetic field and the electron beam direction are perpendicular to the plane of the nanostripe. The symbols $L$ and $P_{SkCh}$ represent the thickness of the nanostripe in the electron beam direction and the period of the single skyrmion chain, respectively. The coordinate system is marked in both panels.

designed to include the helical period $L_D = 70$ nm of bulk FeGe (ref. 25). In this way, a comprehensive $H$-$W_y$ phase diagram for a wide range of nanostripe widths was constructed from a single sample. Figure 2a,e shows bright-field TEM images of the fabricated wedge-shaped nanostripe. The regions marked by two white frames were selected for detailed analysis using off-axis EH at temperatures of 220 and 95 K, respectively. Figure 2b–d,f–k shows quantitative magnetic induction maps of the spin texture measured experimentally in the presence of different external magnetic fields using off-axis EH, with the directions and magnitudes of the projected in-plane magnetization fields shown in the form of composite contour-colour maps. It should be noted that the recorded magnetic signal extends to the very edges of the nanostripe and is not affected by imaging artifacts, uniquely providing the opportunity to test the predicted existence of

twisted edge states[14–17]. The imaging conditions and analysis procedure are described in Supplementary Figs 3 and 4.

When the nanostripe was zero-field-cooled from room temperature to 220 K (Fig. 2b–d), it was initially found to contain a complicated magnetic ground state comprising a mixture of regular, curved and vortex-like magnetic helices with distorted edge spins (Fig. 2b). When a magnetic field was applied normal to the nanostripe plane, the spin helices in the nanostripe evolved in a complex process to form a single skyrmion chain (SSC) at $\mu_0 H \sim 148$ mT (Fig. 2c). Significantly, the skyrmions can be seen directly to adopt a sequence of compressed, regular and stretched morphologies with increasing nanostripe width, fully consistent with the theoretical prediction shown in Fig. 1a. Similar distorted skyrmions were recently reported in two-dimensional FeGe$_{1-x}$Si$_x$ samples as a result of local lattice disorder around the edge of a crystal grain[24]. The distortions become less pronounced when the applied magnetic field is increased. For example, at $\mu_0 H \sim 222$ mT the skyrmions decrease in size, adjust their positions and adopt more circular shapes (Fig. 2d). Meanwhile, skyrmions in the narrower part of the nanostripe disappear or migrate, while those in the wider part of the nanostripe form a zigzag skyrmion chain (ZSC) (dotted rectangle in Fig. 2d). Notably, the skyrmion state is always accompanied by a complete chiral edge twist, which is observed directly in the magnetic induction maps and is characterized by a single-twist rotation of the magnetization and nearly in-plane spins around the edge of the nanostripe (short white arrows in Fig. 2c)[33]. According to theoretical analysis, such an edge spin configuration can be regarded as a type of surface state in a chiral magnet, which preserves the magnetic chirality of the spin texture around the edge[14–17] and has been anticipated to play a key role in current-induced skyrmion motion in nanostripes[12,13]. Experimentally, such chiral edge twists have been inferred in our previous Lorentz TEM imaging of magnetic structure in both nanostripes[18] and nanodisks[19]. However, the contrast around the edge associated with Fresnel fringes from the rapid change in specimen thickness, as discussed in the introduction, complicated the observed images, especially for sample sizes of below 100 nm. Here, we give the first clear and convincing images of the edge magnetic state of a FeGe nanostripe.

The elliptically distorted skyrmions also persist at lower temperature. However, their formation process is significantly different from that at higher temperature. Previous investigations have established that low temperatures are not beneficial for the formation of skyrmions, with the helix-to-skyrmion transformation depending on the initial helical state[18,25]. After applying the same zero-field cooling procedure to 95 K, the nanostripe contains a mixed helical state, whose wave vector **k** is parallel and perpendicular to its long axis in the narrower and wider part of the nanostripe, respectively (Fig. 2f). At $\mu_0 H \sim 217$ mT, the helices transform into a SSC state only where **k** is parallel to the long axis (Fig. 2g). In this case, the helix-to-skyrmion transformation has a precise one-to-one correspondence[18], with each helix corresponding to a single skyrmion, as indicated by the curved dotted white lines in Fig. 2f–h. This special helix-to-skyrmion transformation has been observed in nanostripes with widths above 130 nm (ref. 18). Here, we confirm that the same behaviour is followed in a much narrower (45–150 nm) nanostripe. At higher magnetic fields, the number of skyrmions remains unchanged and they show a similar behaviour to that at $T \sim 220$ K, with a reduced size and ellipticity (Fig. 2h). Moreover, by applying a cyclical magnetization process at $T \sim 220$ K and then cooling the nanostripe to 95 K in zero field, we were able to control the initial helical state and to ensure that the direction of the wave vector **k** was almost completely along the long axis of the

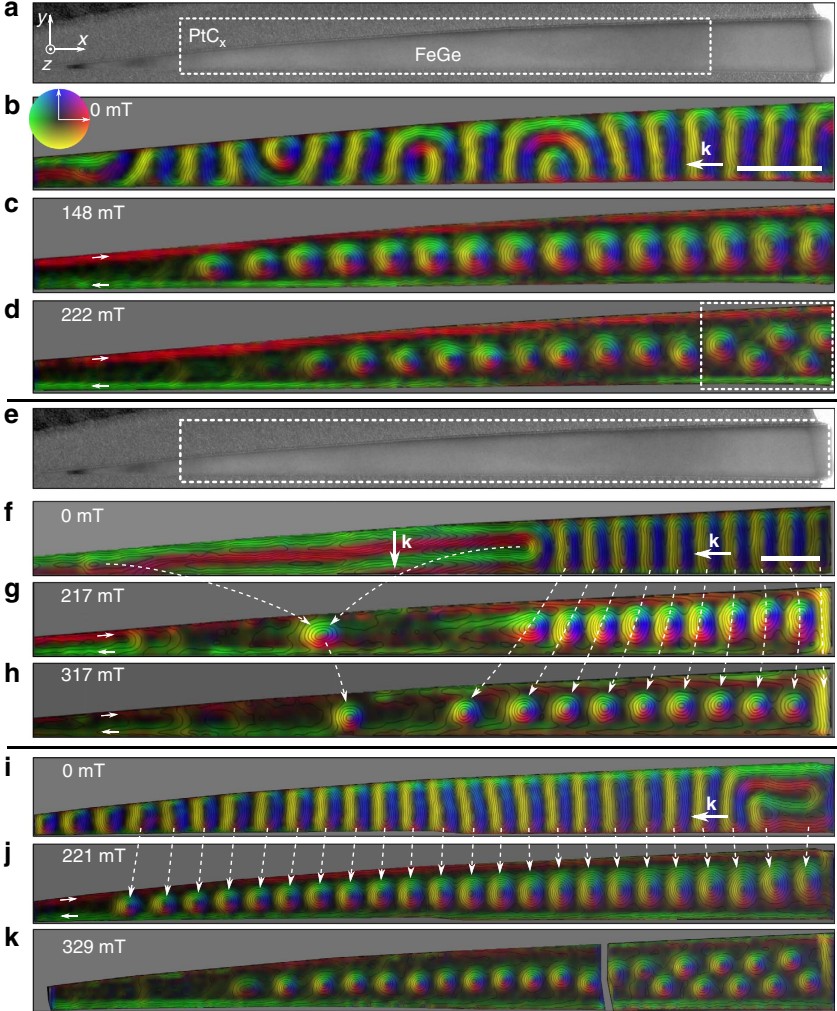

**Figure 2 | Magnetic induction maps recorded from an FeGe nanostripe.** (**a**,**e**) Bright-field TEM image of the wedge-shaped FeGe nanostripe. The white frames show the regions from which magnetic induction maps were recorded using off-axis EH. Experimentally measured magnetic induction maps showing the magnetic field dependence of the spin texture are shown in **b**–**d** for a temperature of 220 K and **f**–**k** for a temperature of 95 K. The direction of the measured in-plane magnetic induction follows the colour wheel shown in **b** (red = right, yellow = down, green = left, blue = up). The short straight white arrows in **c**,**d**,**g**,**h**,**j** show the direction of the spins around the edge of the nanostripe. A zigzag skyrmion chain is marked by a white frame in **d**. A one-to-one correspondence in the helix-to-skyrmion transformation, with each helix corresponding to a skyrmion, is indicated by the curved dotted white lines in **f**–**h** and **i**,**j**. The vectors **k** in **b**,**f**,**i** are defined as the propagation wave vectors of the magnetic helix in zero field. The scale bars in **b**,**f** are both 150 nm.

nanostripe (Fig. 2i). In response to an applied magnetic field, the helix-to-skyrmion transition again followed a one-to-one behaviour, leading to similar skyrmion morphologies (Fig. 2c,j).

The distinctive nucleation process that we observe for magnetic skyrmions at low temperature from a helical spin spiral is reproduced in atomistic simulations based on direct energy minimization (see the Methods/Numerical simulations section) for a nanostripe of fixed width at zero temperature (Supplementary Fig. 5). Notably, both the simulation and the experimental results indicate that the period of the spin spiral remains almost unchanged over the complete range of their existence. This finding is remarkably different from observations made on spatially extended thin films and bulk crystals[34], in which the helical period increases monotonically with applied field. The reason for this difference is that the adjustment of the spin spiral period to an equilibrium value requires an unwinding of the twisted edge state and is associated with a high energy barrier. The resulting one-to-one helix-to-skyrmion transition

means that the period of the skyrmion chain inherits the periodicity of the parent spin spiral, resulting in a difference between the period of a single skyrmion chain $P_{SkCh}$ and the equilibrium distance $P_{SkL}$ in a skyrmion lattice in an extended film (Supplementary Fig. 5).

We further quantified the effects of confinement on skyrmion morphology using both our experimental results and a theoretical model. The results obtained at $T \sim 220$ K were used to understand the influence of nanostripe width on skyrmion morphology and to build a width-field phase diagram (Fig. 4). Figure 3a,b show a selection of experimental off-axis EH images recorded at $T \sim 220$ K and $\mu_0 H \sim 148$ mT. The deformation of each skyrmion was well estimated by fitting an elliptical shape with semi-axes $a$ and $b$ to its magnetic contrast profile obtained using off-axis EH. Recent experimental studies using differential phase contrast (DPC) imaging performed in a scanning TEM revealed an intrinsic six-fold symmetry of the internal structure of skyrmion lattice cells in two-dimensional FeGe crystals[35]. We have also

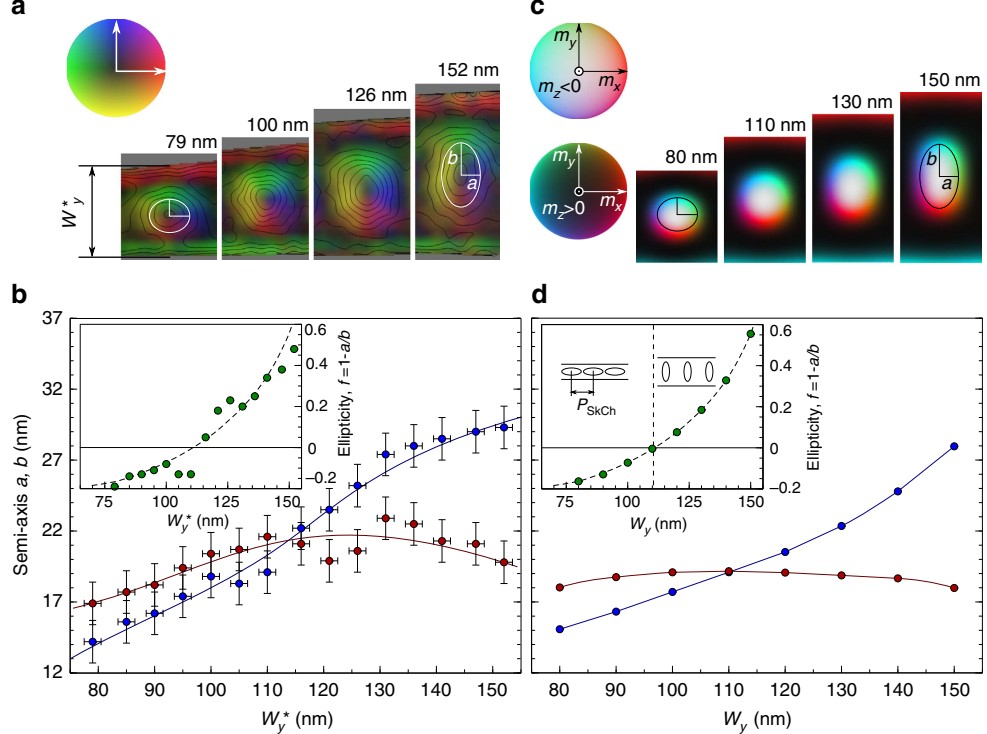

**Figure 3 | Measurements of skyrmion morphology.** (**a**) Selection of experimentally measured magnetic induction maps of distorted skyrmions recorded in an applied magnetic field $\mu_0 H \sim 148$ mT, shown for different values of nanostripe width. The colour wheel has the same definition as in Fig. 2b. (**b**) Experimentally measured values of $a$ and $b$ determined from phase images of skyrmions and plotted as a function of the position-dependent width, $W_y^*$ along the wedge-shaped nanostripe. The data points are shown as symbols with thin solid lines plotted to guide the eye. The inset shows the experimentally measured ellipticity (solid dots) displayed together with a theoretical prediction. The error bars are determined from the phase resolution of electron holographic imaging. (**c**) Magnetization maps determined from simulations of skyrmions in nanostripes of different width. (**d**) Semi-axes $a$ and $b$ determined from the simulations and plotted as a function of nanostripe width. The inset shows the ellipticity measured from the simulations.

observed this hexagonal skyrmion structure in two-dimensional FeGe using off-axis EH[36]. Such observations confirm that the shapes of isolated skyrmions can be modified significantly in confined geometries. Here, we describe the skyrmion ellipticity by defining a parameter $f = 1 - a/b$, which is zero when a skyrmion is circular and takes negative and positive values when it has longitudinal and transverse ellipticity in the $x$ direction, respectively. In the experimental plot shown in Fig. 3b, the semi-axis $a$ exhibits a non-monotonic dependence on nanostripe width, while the semi-axis $b$ increases continuously with nanostripe width and is almost twice as large in the wider part of the wedge as in the narrower part. These results are reproduced in a theoretical calculation by means of a similar analysis, with the critical nanostripe width $W_y^c$ at which $a = b$ determined to be $\sim 110$ nm. It should be noted that the formation of elliptical skyrmions in a strongly confined geometry is distinctly different from the formation of anisotropic distorted skyrmions in a two-dimensional FeGe sample in the presence of a strain-induced modification of the intrinsic magnetic interactions[37].

**Width-field magnetic phase diagram.** Based on all of our experimental results (Supplementary Figs 6 and 7), we constructed a width-field ($W_y^*$-$H$) magnetic phase diagram for the FeGe nanostripe (Fig. 4a). At a low value of applied magnetic field, a distorted helical spin spiral appears. It then transforms into a pure edge twist, a single skyrmion chain or a zigzag skyrmion chain in an applied magnetic field of $\sim 75$ mT, depending on the width of the nanostripe $W_y^*$. We identified a limiting range

of nanostripe widths between 79 and 140 nm, within which a single skyrmion chain is supported. For $W_y^* < 79$ nm, no complete skyrmions survive, while for $W_y^* > 140$ nm either a SSC or a ZSC forms, depending on the applied field. The appearance of a ZSC reflects the tendency of interacting skyrmions to condense into a hexagonal lattice when they are densely packed[8]. In the narrow part of the nanostripe, at $\mu_0 H \sim 105$ mT the helix-to-skyrmion transition is characterized by the formation of an incomplete skyrmion, with a half-skyrmion attached to one edge and a twisted state on the other edge (small magenta domain in Fig. 4a; see also Supplementary Fig. 6b). In Supplementary Fig. 5d, we show qualitatively similar incomplete skyrmions obtained during energy minimization using the NCG method. However, it has to be mentioned that, strictly speaking, such magnetization states obtained using a direct minimization method are not a true physical realization and only final magnetization configurations corresponding to the equilibrium state can be compared to the experimental data. At higher magnetic fields, the skyrmions gradually lose their stability and collapse into a conical or field-saturated state. We estimate an optimal width for the nanostripe of $\sim 110$ nm, which corresponds to $W_y \approx 1.6 L_D$. For a nanostripe with $W_y < 1.6 L_D$, the range of existence of skyrmions in an applied magnetic field is significantly lower than for skyrmions in an extended film.

**Discussion**

Our experimental phase diagram agrees closely with a corresponding theoretical phase diagram (Fig. 4b), in which transitions between magnetic phases were determined by comparing their

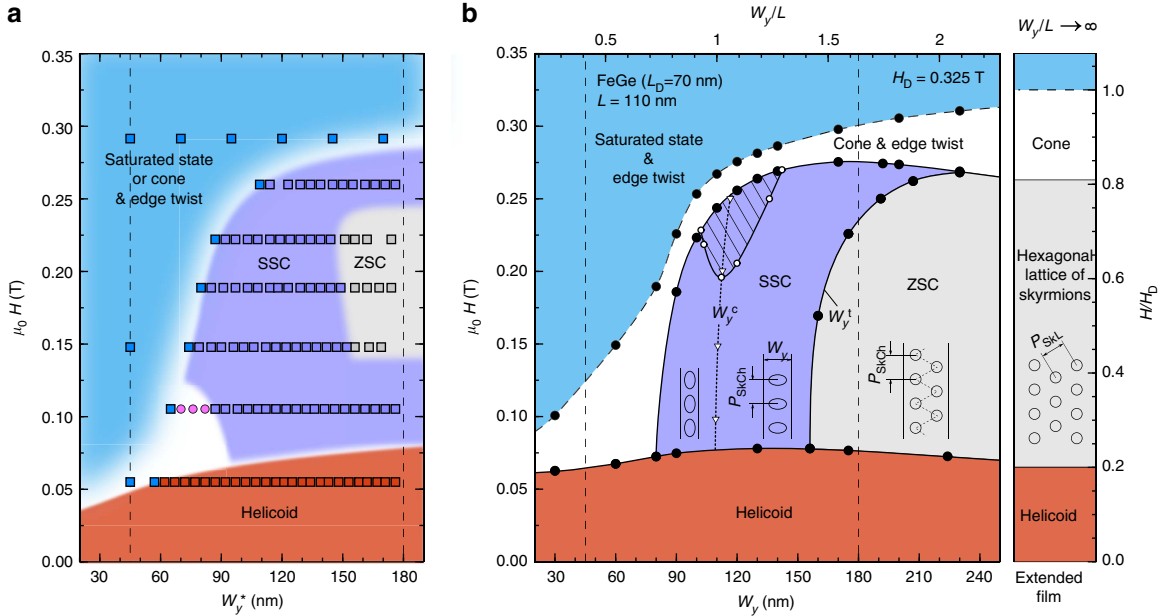

**Figure 4 | Width-field magnetic phase diagram in a nanostripe.** (**a**) Experimentally observed magnetic states in the wedge-shaped nanostripe in an applied magnetic field and at a fixed temperature $T = 220$ K The filled squares represent experimental data. Orange, purple, grey and blue squares correspond to helical spiral, single skyrmion chain (SSC), zigzag skyrmion chain (ZSC) and field-saturated states, respectively. Magenta circles correspond to an intermediate metastable state of half skyrmions attached to the edge. Transition lines between domains of corresponding colour are used to guide the eye. (**b**) Theoretical phase diagram of equilibrium magnetic states calculated for an infinitely long FeGe nanostripe of width $W_y$ and fixed thickness $L = 110$ nm in an applied magnetic field $H$. The filled circles represent calculated points. The colour notations are the same as in **a**. The phase diagram is extended for an infinite film ($W_y/L \to \infty$) of corresponding thickness. The upper and right axes are equivalent to the lower and left axes, respectively and are given in reduced units of thickness $L$ and saturation field $H_D$. The vertical dashed lines in **a,b**, at 30 and 180 nm, mark the limited width of the wedge in the experimental observation. $W_y^c$ and $W_y^t$ correspond to the critical width for ideally circular skyrmions and the transition from a SSC to a ZSC, respectively. The triangular dots are taken from calculations. The dotted line is included as a guide to the eye. Above a certain magnetic field (marked by empty dots) and assuming a fixed skyrmion density, the skyrmions become circular over a range of nanostripe widths, as marked by the dashed region. $P_{SkL}$ represents the period of the skyrmion lattice. The dashed region in **b** corresponds to a circular skyrmion.

energies (see the Methods/Numerical simulations section). The width of the nanostripe was further extended significantly in the simulations. Slight discrepancies between the experimental and theoretical phase diagrams arise from an experimentally observed weakly hysteretic behaviour of the system, which originates from the presence of finite energy barriers between magnetic states. In the theoretical phase diagram, $W_y^c$ corresponds to the critical width for ideally circular skyrmions. Below and above $W_y^c$, the skyrmions always exhibit longitudinal and transverse elliptical distortions, respectively. However, above a certain magnetic field and assuming a fixed skyrmion density, as observed experimentally, the skyrmions become circular over a wide range of nanostripe widths (see the dashed region in Fig. 4b). Outside this range, the skyrmions are always elliptically distorted. However, at high magnetic fields $\mu_0 H \gtrsim 300$ mT ($H \gtrsim 0.9 H_D$ in reduced units) and finite temperatures, their elliptical distortions become negligibly small and in practice indistinguishable on a background of thermal fluctuations for any nanostripe width (see, e.g., Fig. 2d,h,k).

The good agreement between our experimental and theoretical results allows us to obtain further insight into the evolution of elliptical skyrmions and the nucleation process by calculating the equilibrium periods of helicoids and skyrmions as a function of applied magnetic field (Fig. 5). For an infinitely wide film without edge effects, the helicoid remains metastable, with a period that increases with applied magnetic field up to very high field of $\sim 210$ mT (red solid circles in Fig. 5)[38]. In contrast, for the nanostripe, in the presence of edges the period of the helicoid

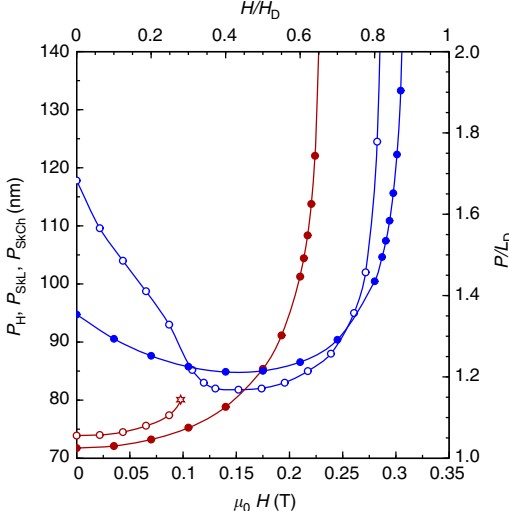

**Figure 5 | Equilibrium period of a helicoid and a skyrmion chain.** $P_H$ is the calculated period of a helicoid. The red and blue dots correspond to the helicoid and skyrmion, respectively. The filled circles correspond to an infinitely wide film without edge effects, for which the helicoid remains metastable up to $\sim 210$ mT. The open circles correspond to a nanostripe, for which the period of the helicoid increases up to a field of $\sim 100$ mT. The red star represents the critical magnetic field, above which a helicoid that is confined to a nanostripe loses its stability.

increases with increasing magnetic field up to $\sim 100\,\text{mT}$ (red hollow circles in Fig. 5), above which it loses its stability. Above a critical field (marked by a star), there are no minima on the energy landscape corresponding to a helicoid. The helicoid then transforms into a single skyrmion chain or a zigzag skyrmion chain, depending on the nanostripe width (Fig. 4b). It should be noted that the transition field of $\sim 70\,\text{mT}$ in the phase diagram (Fig. 4b) is the field at which the energies of the spin spiral and the skyrmion chain are equivalent, whereas the instability of the helicoid appears at $\sim 100\,\text{mT}$. At a finite temperature, the experimentally observed transition is expected to appear between these two critical fields.

Our results are based on the application of state-of-the-art electron holography in combination with advanced computer simulations based on atomistic models and direct energy minimization. In this context, our experimental results could not have been obtained using other methods such as small angle neutron scattering[4], transport measurements[39] or Lorentz TEM[6]. From a methodological perspective, the experimental approach that we use offers the prospect of allowing two and three-dimensional magnetic textures in other nanometer-scale spin systems to be quantified[40] in future studies. Such a capability is currently lacking using any other technique. In a broader context, our experimental results and theoretical analysis therefore open avenues for exploring quantum confinement in other complex non-linear spin systems.

Our results can be envisioned for several potential applications. First, the high flexibility of spatially-confined magnetic skyrmions that we observe allows them to adapt their shape and size in a nanostructure whose size is comparable to or even smaller than the size of an equilibrium skyrmion in an extended film. For nanostripe widths $W_y < W_y^c$, such skyrmions remain stable even at a relatively high temperature of $T \sim 0.8\ T_c$. Since the efficient coupling of electron currents to skyrmions is related to topological charge and is independent of skyrmion morphology[9], we expect that elliptical skyrmions can be moved as efficiently as circular skyrmions. Moreover, in contrast to the so-called precession and breathing modes that are exhibited by approximate circular skyrmions in extended films[41], we predict unique oscillating and stretch-out excitation modes in alternating magnetic fields for elliptically-distorted skyrmions in nanostripes whose widths are smaller than twice the period of an equilibrium spin spiral, that is, $W_y < 2L_D$. A further important aspect revealed by our observations is the direct visualization of a mechanism of skyrmion nucleation that is almost independent of their initial state and characterized by the conservation of skyrmion number with respect to the number of helical spirals in the nanostripe. In contrast to earlier proposals[12,21,42], the controlled formation of a well-defined number of skyrmions in a nanostripe whose width lies in the range, $L_D < W_y < 2L_D$ can be achieved simply by varying the applied magnetic field. Such an approach provides a simple and efficient method of skyrmion nucleation that is highly suitable for applications in spintronic devices. For instance, by applying a magnetic field locally to a nanostripe with a fixed length equal to an integer number of spin spiral periods $W_x = N \times L_D$, one can reliably initiate the nucleation of $N$ skyrmions, which then can be pushed onto a skyrmion track by applying an electric current. Moreover, the range of nanostripe widths $L_D < W_y < 2L_D$ can be considered as a range for the width of a skyrmion racetrack device. Indeed, according to our observations and theoretical modelling, for $W_y$ outside this range we observe an instability of a single skyrmion chain in the form of a collapse of individual skyrmions when $W_y \ll L_D$ or an instability in the form of the formation of a zigzag chain of skyrmions when $W_y \gg 2L_D$.

## Methods

**Specimen preparation.** Polycrystalline B20-type FeGe samples were synthesized using a cubic anvil-type high-pressure apparatus[18]. Structural characterization by X-ray diffraction and susceptibility measurements were used to confirm the crystalline quality of the bulk FeGe. A wedge-shaped nanostripe was prepared for TEM observation using a lift-out method in a focused ion beam (FIB) scanning electron microscope (SEM) dual beam workstation (FEI Helios Nanolab 600i) equipped with a gas injection system (GIS) and a micromanipulator (Oxford Omniprobe 200+). Details of the sample fabrication procedure are given in Supplementary Fig. 1.

**Off-axis electron holography.** Transmission electron microscopy was carried out in an FEI Titan 80-300 XFEG TEM operated at 300 kV. For magnetic imaging, the specimen was placed in magnetic field-free conditions (Lorentz mode) with the conventional objective lens turned off. The excitation of the objective lens was varied to apply magnetic fields normal to the specimen plane over a field range of between 0 and $\sim 1.5\,\text{T}$. A double tilt liquid nitrogen specimen holder (model 636, Gatan Co.) and a temperature controller were used to vary the specimen temperature between 380 and 95 K. In off-axis electron holography experiments, the biprism voltage was typically set to 90 V to produce an overlap interference width of 1,200 nm and a holographic interference fringe spacing of 3.6 nm at a nominal magnification of 38,000. For hologram recording, a cumulative acquisition approach was used to record 20 holograms (with an exposure time of 6 s for each hologram). Off-axis electron holograms were reconstructed numerically using a standard Fourier transform based method with sideband filtering using custom-designed Matlab codes. Contour lines and colour maps were generated from recorded magnetic phase images to yield magnetic induction maps (see Supplementary Figs 3 and 4 for details).

**Numerical simulations.** Theoretical models of skyrmions were constructed using a classical spin Hamiltonian for a simple cubic lattice consisting of only three energy terms: the Heisenberg exchange interaction, the Dzyaloshinskii–Moriya interaction (DMI) and the energy of the applied magnetic field[32,43] in the form:

$$E = -J \sum_{\langle ij \rangle} \mathbf{m}_i \cdot \mathbf{m}_j - \sum_{\langle ij \rangle} \mathbf{D}_{ij} \left[ \mathbf{m}_i \times \mathbf{m}_j \right] - \mu_0 \mathbf{H} \sum_i \mathbf{m}_i, \tag{1}$$

where $J$ is the exchange coupling constant, $\mathbf{D}_{ij}$ is the Dzyaloshinskii–Moriya vector defined as $\mathbf{D}_{ij} = D \cdot \mathbf{r}_{ij}$, $D$ is the DMI scale constant, $\mathbf{r}_{ij}$ is a unit vector pointing from lattice site $i$ to lattice site $j$, $\mu_0$ is the vacuum permeability, $\mathbf{H}$ is the external magnetic field and $\mathbf{m}_i = \mathbf{M}_i/M_i$ is a unit vector of the magnetic moment at lattice site $i$. The symbol $\langle ij \rangle$ denotes summation over all nearest-neighbor pairs. Heisenberg exchange and DMI are assumed to be isotropic in all three spatial directions. Following a standard approach, we ignore the contribution of magnetocrystalline anisotropy[32], which in the case of FeGe is negligibly small with respect to other energy terms, especially at relatively large temperatures as in our experimental setup[44]. Because of mainly Bloch-type modulations of the magnetization in FeGe, the stray field effect can be described exclusively in terms of so-called surface magnetostatic charges[7], which in the case of a nanostripe produce a closed flux of the stray field mainly outside the sample. Therefore, we expect a relatively small contribution from magnetostatic interactions and in the first approximation we neglect it. Nevertheless, in the most general case and for a precise quantitative description of the system the contribution of stray field interactions should be taken into account.

Following earlier studies[8,32], we introduce the notation

$$L_D = 4\pi \mathcal{A}/\mathcal{D} = 2\pi a J/D \tag{2}$$

$$H_D = \mathcal{D}^2/(2M_s \mathcal{A}) = D^2/(J\mu_0) \tag{3}$$

where $L_D$ is the period of a homogeneous helical spin spiral in zero magnetic field, $H_D$ is the saturation magnetic field corresponding to the transition between the conical phase and a field saturated ferromagnetic state for the bulk crystal, $\mathcal{A}$ and $\mathcal{D}$ are micromagnetic constants for the exchange interaction and the DMI, respectively, $M_s$ is the magnetization of the material and $a$ is the lattice constant.

In the present simulations, we used $L_D = 70\ a$ and a size for the simulated domain of between 30 and 512 spins in all three spatial directions. A value for $H_D$ of 0.325 T was chosen based on both the present results and previous experimental measurements[45]. The realistically large $L_D/a$ ratio and the choice of value for $H_D$ resulted in good qualitative and quantitative agreement between the simulations and the experimental measurements. Taking into account the expression for $L_D$ in equation (2) in the calculations, we set the coupling constants in arbitrary units of energy to be, $D/J = 2\pi/70 \approx 0.09$ and the applied field $H$ in units of saturation field $H_D$ according to equation (3).

For direct minimization of the model Hamiltonian (equation (1)), we used a non-linear conjugate gradient (NCG) method employing adaptive stereographic projections for the magnetization vectors, all implemented on NVIDIA CUDA architecture[32]. As an initial state for the helicoid and conical phases, we used an ordinary Bloch-type spiral with a propagation vector along the long side of the nanostripe ($x$-axis) or perpendicular to it and along the applied field ($z$-axis), respectively. For the skyrmion phase, we used as the initial configuration a simple ferromagnetic state with its magnetization along the direction of the applied field

(positive direction of the z-axis) and cylindrical domains with opposite magnetization. At each point on the phase diagram of magnetic states shown in Fig. 4, we identified the equilibrium period of the spirals and the equilibrium density of skyrmions in the system, which correspond to the lowest energy density of the particular phase and then compared the energies of all competing phases. Moreover, an NCG method was used to simulate the field-driven magnetization process of skyrmion formation from the helical state. In this case, the initial magnetic state was set to be spin helices and the external magnetic field was then increased gradually so that a metastable state with a local minimum energy was obtained (Supplementary Fig. 5). This procedure is different from a conventional Landau–Lifshitz–Gilbert (LLG) dynamics simulation[46], in which the magnetization process is obtained by solving the LLG equation. However, in a certain sense, the NCG method can be referred to as pseudo-dynamic, meaning that at each iteration step it follows the direction of the lowest energy gradient, but without a pre-defined time step.

**Data availability**. All relevant data are available from the authors on reasonable request. See author contributions for enquiries about specific data sets.

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

## Acknowledgements

This work was supported by the Natural Science Foundation of China, Grant Nos. 51622105, 11474290, 11374302 and U1432251; the Key Research Program of Frontier Sciences, CASOQYZDB-SSW-SLH009; the Youth Innovation Promotion Association CAS No. 2015267; the CAS/SAFEA international partnership program for creative research teams of China and the National Key Basic Research of China. Work by F.N.R. was carried out within the state assignment of FASO of Russia (theme Quantum No. 01201463332). The research leading to these results has received funding from the European Research Council under the European Union's Seventh Framework Programme (FP7/2007-2013)/ERC grant agreement number 320832.

## Author contributions

H.D., C.J. and M.T. conceived the project and prepared the samples. Z.-A.L. performed the electron holography experiments and analysed the holographic data with assistance from A.K., J.C. and F.Z. S.K. and F.N.R. developed the model calculations. The manuscript was prepared by H.D., Z.-A.L. and N.S.K., with contributions from M.F., R.E.D.-B., S.B. and Y.Z. All authors discussed the results and contributed to the manuscript.

## Additional information

**Publisher's note**: 

