## [Peer review file · Nature Communications]

Reviewers' comments:

Reviewer #1 (Remarks to the Author):

Authors study the magnetisation configurations in wedge-shaped FeGe nanowires using both experimental and simulation approaches. They demonstrate the ability of skyrmions to adapt their shape (transversal, circular, longitudinal) according to the nanowire width. In addition, they find geometry and external magnetic field conditions under which skyrmion chain emerges. The manuscript is reasonably well written, technically sound, with several misspellings and missing references.

Remarks:

1. I find the results interesting, but I have several concerns that are related to the novelty and importance of the submitted manuscript. More specifically, a simple search for skyrmions in confined geometries results in several already published works. Some of them are:

[1] H. Du, W. Ning, M. Tian, and Y. Zhang, Phys. Rev. B 87, 14401 (2013).

[2] H. Du, W. Ning, M. Tian, and Y. Zhang, EPL 101, 37001 (2013).

[3] J. Sampaio, V. Cros, S. Rohart, A. Thiaville, and A. Fert, Nat. Nanotechnol. 8, 839 (2013).

[4] S. Rohart and A. Thiaville, Phys. Rev. B 88, 184422 (2013).

[5] M. Beg, R. Carey, W. Wang, D. Cortés-Ortuño, M. Vousden, M.-A. Bisotti, M. Albert, D. Chernyshenko, O. Hovorka, R. L. Stamps, and H. Fangohr, Sci. Rep. 5, 17137 (2015).

[6] H. Du, R. Che, L. Kong, X. Zhao, C. Jin, C. Wang, J. Yang, W. Ning, R. Li, C. Jin, X. Chen, J. Zang, Y. Zhang, and M. Tian, Nat. Commun. 6, 8504 (2015).

[7] X. Zhao, C. Jin, C. Wang, H. Du, J. Zang, M. Tian, R. Che, and Y. Zhang, Proc. Natl. Acad. Sci. 113, 4918 (2016).

Although authors cite some of these references, it is not clearly explained how the findings in submitted manuscript build on already published work, especially reference 6, and what is the novelty in comparison to them. Manuscript would benefit from providing a detailed overview of already existing literature which would also emphasise the novelty of this work.

2. Authors claim that this work is the first observation of edge spin configuration. However, it seems the magnetisation tilting at the sample edges was already reported in Ref. 6 and 7 and analytically predicted in Ref. 4. Do authors have any arguments on that?

3. Regarding methodology, authors use discrete spin lattice model containing only ferromagnetic exchange, Dzyaloshinskii-Moriya, and Zeeman energy terms. They minimise the system's energy using conjugate gradient method. However, at several places in the main text and supplementary information they refer to "dynamical simulations" and show intermediate states (Supplementary Fig. d and e). It is important to emphasise that the "path" system takes to minimise its energy in conjugate gradient method does not correspond to the real path in the energy landscape if simulated using Landau-Lifshitz-Gilbert dynamics. Therefore, claiming that these states are intermediate states or that dynamic simulations are used is not justified.

4. In order to allow reproducibility of simulation results, J and D parameters should be stated. Also, omitting magnetocrystalline anisotropy and dipolar interactions in FeGe needs

justification.

5. Authors state that "transitions between magnetic phases were determined by comparing their energies", but no explanation on how the system was initialised (what states were relaxed) is given. At zero temperature, initial state used in the conjugate gradient method is crucial.

6. Authors make several too optimistic and in my opinion not justified claims. In abstract they state that "The flexibility and ease of formation of geometrically confined magnetic skyrmions that we describe has immediate implications for the design of skyrmion-based memory devices". Also, they claim that "the controlled formation of a well-defined number of skyrmions in a nanostripe whose width lies in the range, $L_d < W_y < 2L_d$, can be achieved simply by varying the applied magnetic field". These claims need more detailed explanation, especially how "controlled formation of a well-defined number of skyrmions" can be practically achieved.

In summary, although there are no major flaws that would compromise the reported results, the manuscript needs a more detailed overview of already published literature and details on the simulation method.

Reviewer #2 (Remarks to the Author):

Review for manuscript NCOMMS-16-18538-T

What are the major claims of the paper?

The authors have imaged the magnetic induction from skyrmions in a nanostripe of FeGe, under different conditions, showing that these will take on elliptical form in a predictable manner depending on the applied field and width of the nanostripe. From these images they create a magnetic phase diagram, which is in agreement with a theoretically calculated phase diagram for the experimentally realized conditions. This agreement allows the authors to use their theoretical calculation to predict the states in a much wider range of fields and stripe widths.

Furthermore, the authors show clear and convincing images of the edge magnetic state of the FeGe nanostripe, using the off-axis holography TEM technique, confirming previous suggestions of a twisted edge state. They also show a clear mechanism for the nucleation of skyrmions from helical states in the nanostripe. This control over the formation of skyrmions is vital, if they are to be used for technological applications.

Are they novel and will they be of interest to others in the community and the wider field?

The clear and convincing imaging of ellipticity, skyrmion formation and magnetic edge states are novel and likely to be of great interest to the skyrmion community. The ability to control the formation of skyrmions is vital for proposed technological applications.

Is the work convincing, and if not, what further evidence would be required to strengthen the conclusions?

The authors are to be commended on their excellent description of the experimental method of electron holography and the clear explanation of how they arrived at the images of magnetic induction shown in the paper through subtraction of the electrostatic potential. This detailed description of the experimental methods (appropriately in the supplementary information) help make the data very convincing. The clear magnetic induction maps, and the comparisons with theoretical calculations, make the conclusions very convincing.

On a more subjective note, do you feel that the paper will influence thinking in the field? I believe the discussion of ellipticity of the skyrmions, and the possibility of these elliptical states having different excitation modes to circular skyrmions in response to alternating fields are likely to give rise to further study. The control of skyrmion formation is a key technological aspect, and other researchers can reproduce it in other samples it is an important step towards applications.

Please feel free to raise any further questions and concerns about the paper.

There are a number of instances of "Error! Reference source not found" in the paper where references should be; these should be checked and the correct reference number and information inserted.

The parameter W_{yc} is not explicitly defined when it is first presented on page 5; from the context this seems to be a critical wedge width for circular skyrmion formation, but it would improve clarity to have this explicitly stated at this point.

In figure 3b and d the experimental and simulated values of long and short axes are shown,

presumably as the data points. There are also lines in these figures; the authors should explain what these lines are. This is especially important for figure 3b, where the lines do not go through the data points, so presumably represent some sort of fit.

An explicit reference to figure 4 (the width-field magnetic phase diagram) on line 178 of page 8 should be included; otherwise there is only one parenthetical reference to figure 4a, which is one of the main findings of the paper.

We would also be grateful if you could comment on the appropriateness and validity of any statistical analysis, as well the ability of a researcher to reproduce the work, given the level of detail provided.

I believe the level of detail provided regarding the electron holograph acquisition and data analysis is sufficient for a researcher with the appropriate equipment and knowledge of the techniques to be able to reproduce the work. The equations and materials parameters used for the theoretical calculations should also provide enough information for someone to reproduce these results.

In conclusion, I recommend that this manuscript is accepted for publication, after the minor changes mentioned within this report.

Reviewer #3 (Remarks to the Author):

The paper from Jin and co-workers reports the observation of elliptical distortions of individual skyrmions in a nonassertive of FeGe. The experiments are carried out via state-of-the-art electron holography in a TEM.

On the technical aspects of the manuscript, I have minor objections. I found the data beautiful and their interpretation mostly believable. The authors observe very clearly the deformations of the skyrmions and produce a phase diagram for the nanostripe by investigating a large parameter space of temperatures and magnetic fields.

The results are well described and presented, with high quality and clear images. Scientifically, the observation of these distortions is of great value for the description of topological magnets, and so is the observation of the chiral edge twist reported here for the first time.

I have some criticism on the following aspects of the paper:

- the authors put a lot of emphasis on the impact that these results may have for applications in data storage devices. While I agree that this kind of research is intended to seed new ideas in spintronic devices, one should be careful in not overselling rather fundamental research for non-realistic applications. I believe that the authors should make an effort in the introduction to explain a bit better to the general reader what really is the relevance of this contribution for applications. For example, they mention passing by that the race-track memory method could benefit from nanostripes supporting skyrmions. A little explanation on why and how would not harm here.

-Both in the abstract and the introduction, the phrase "width in the range of only a few tens of nanometers" is used. In spoken English, the phrase "a few" is conventionally taken to mean "three". The width of the regions analysed in the range 45-150nm, much greater than a few tens!

-At the bottom of page 3 the authors aim to put their electron holography measurements into context by mentioning Fresnel imaging (which has been much used to image skyrmions and references should be given). The neglect to mention that the Lorentz STEM technique of DPC has been applied to discover aspects of skyrmion structure with nanometer spatial resolution. Such as in:

1) McGrouther et al., arXiv:1606.04681v2

2) Matsumoto et al, Sci. Adv. 2016, 2:e1501280

Both articles show the possibility for distortions of skyrmion structure, in particular the latter, which highlights heavily distorted skyrmions at a crystal grain chirality boundary, similar to states shown by the authors.

-Later, at the top of page 5, the authors over-generalise their

description of chiral magnetic phenomena. When describing the behaviour in bulk crystals they completely neglect to mention the existence of the A-phase pocket. When describing behaviour in thin crystal sections, care is needed to not refer to their sample as a film. Also, their description of the form of the skyrmion in the lattice contradicts that shown by McGrouther et al., but also one of their own works, arXiv:1606.05723v1, which shows that the skyrmions possess hexagonal not circular character.

Of course the case might be different for isolated skyrmions and those shown by the authors when they are not organized in a triangular lattice. However, the electron holography images appear far too heavily filtered to be able to conclusively make a confident judgement in this case.

- Some references have not compiled in the pdf file, just run a check of the bibliography.

Overall, I find this paper of good quality. The data are obtained with a state of the art TEM for electron holography and they address an issue of current interest in condensed matter physics. I am not that sure about the interest of such a report for an audience that goes beyond the community working on magnetic skyrmions. If the authors can provide a clearer motivation for such a broad interest and other reviewers, not directly working in the field of magnetic skyrmions imaging, feel that this article deserves the attention of a broader community of physicists I would have no further objections once the above has been taken care of.

Response to Reviewer #1

Authors study the magnetisation configurations in wedge-shaped FeGe nanowires using both experimental and simulation approaches. They demonstrate the ability of skyrmions to adapt their shape (transversal, circular, longitudinal) according to the nanowire width. In addition, they find geometry and external magnetic field conditions under which skyrmion chain emerges. The manuscript is reasonably well written, technically sound, with several misspellings and missing references.

Our response: We appreciate the referee's positive assessment of our work. We have corrected the misspellings and added the missing references.

Question 1:

I find the results interesting, but I have several concerns that are related to the novelty and importance of the submitted manuscript. More specifically, a simple search for skyrmions in confined geometries results in several already published works. Some of them are:

[1] H. Du, W. Ning, M. Tian, and Y. Zhang, Phys. Rev. B 87, 14401 (2013).

[2] H. Du, W. Ning, M. Tian, and Y. Zhang, EPL 101, 37001 (2013).

[3] J. Sampaio, V. Cros, S. Rohart, A. Thiaville, and A. Fert, Nat. Nanotechnol. 8, 839 (2013).

[4] S. Rohart and A. Thiaville, Phys. Rev. B 88, 184422 (2013).

[5] M. Beg, R. Carey, W. Wang, D. Cortés-Ortuño, M. Vousden, M.-A. Bisotti, M. Albert, D. Chernyshenko, O. Hovorka, R. L. Stamps, and H. Fangohr, Sci. Rep. 5, 17137 (2015).

[6] H. Du, R. Che, L. Kong, X. Zhao, C. Jin, C. Wang, J. Yang, W. Ning, R. Li, C. Jin, X. Chen, J. Zang, Y. Zhang, and M. Tian, Nat. Commun. 6, 8504 (2015).

[7] X. Zhao, C. Jin, C. Wang, H. Du, J. Zang, M. Tian, R. Che, and Y. Zhang, Proc. Natl. Acad. Sci. 113, 4918 (2016).

Although authors cite some of these references, it is not clearly explained how the findings in submitted manuscript build on already published work, especially reference 6, and what is the novelty in comparison to them. Manuscript would benefit from providing a detailed overview of already existing literature which would also emphasize the novelty of this work.

Our response:

We thank the reviewer for the careful consideration of our work, for the judgment that our results are interesting and for the suggestion that the “*Manuscript would benefit from providing a detailed overview of already existing literature which would also emphasize the novelty of this work.*” In response, we have clarified the novelty of our work and provided a more detailed overview of the published literature on skyrmions in confined geometries by making the following changes to the text:

In the abstract, we have deleted the first sentence in the original manuscript, which read “*Topologically non-trivial objects can change their shape without changing their topological class and cannot easily be created by the smooth deformation of other objects that have a different topological class. Small magnetic whirls in chiral magnets, which are termed magnetic skyrmions, possess non-trivial magnetization topologies due to the peculiar twists of the spins within them.*” Instead, we have written:

“The ability to controllably manipulate magnetic skyrmions, small magnetic whirls with particle-like properties, in nanostructured elements is a prerequisite for a new class of spintronic devices.”

In the introduction, we have added a detailed overview of the published literature on skyrmions in confined geometries and cited all of the literature recommended by the reviewer. On page 3, in order to emphasize the importance and novelty of the present work, we have written:

“In particular, skyrmions are promising candidates for applications in novel data storage devices based on the race-track memory concept¹¹. The underlying design of such devices relies on the use of skyrmions as data bit carriers, which move along a ferromagnetic nanostripe that takes on the role of a guiding track. It is therefore important to be able to controllably form and manipulate skyrmions in nanostructured elements. Over the past few years, many theoretical proposals have been put forward to host and create individual skyrmions in confined geometries¹²⁻¹⁷, including the possibility of locally nucleating an isolated skyrmion by using a spin-polarized current and driving its motion using a current-induced torque^{12,13}. On the experimental side, skyrmion chains and skyrmion cluster states have been observed in FeGe nanostripes¹⁸ and nanodisks¹⁹, respectively, using Lorentz transmission electron microscopy (TEM). Moreover, recent work has demonstrated powerful skyrmion-based information storage functionalities²⁰, such as the ability to generate skyrmion bubbles in geometrically confined CoFeB/Ta films using in-plane currents²¹. However, the optimization of information storage requires an optimal width for a nanostripe that hosts a chain of skyrmions, as well as precise control of their morphology²² and formation in such a confined geometry. Therefore, the next important step is to develop approaches that can be used to control the fine structures of individual skyrmions and to nucleate them in a simple manner in a nanostripe, whose size is similar to that of the skyrmions themselves.”

Question 2:

Authors claim that this work is the first observation of edge spin configuration. However, it seems the magnetization tilting at the sample edges was already reported in Ref. 6 and 7 and analytically predicted in Ref. 4. Do authors have any arguments on that?

Our Response:

The magnetization tilting at sample edges was indeed reported in our previous work Refs. 6 and 7 and analytically predicted in Ref. 4. However, the previous experimental results were obtained in the nanostructures with a width larger than 130 nm using Lorentz TEM. As discussed in the second paragraph of the text (on page 4), the Fresnel (out-of-focus) imaging mode of Lorentz TEM is not well suited to the accurate analysis of magnetic structures near to specimen edges due to the presence of strong Fresnel fringe contrast associated with the rapid change in specimen thickness. Accordingly, Lorentz TEM cannot be used to characterize skyrmions in nanostructures that have sizes of below 100 nm. In contrast, the technique of off-axis electron holography in the TEM, which we apply in the present manuscript, has the ability to image sub-100-nm magnetic structures and magnetic edge states directly. In order to avoid confusion, we have deleted the sentence “*Our results give the first clear observation of this magnetic state*”.

In the revised manuscript, we have written in pages 7 and 8:

“Experimentally, such chiral edge twists have been inferred in our previous Lorentz TEM imaging of magnetic structure in both nanostructures¹⁸ and nanodisks¹⁹. However, the contrast around the edge associated with Fresnel fringes from the rapid change in specimen thickness, as discussed in the introduction, complicated the observed images, especially for sample sizes of below 100 nm. Here, we give the first clear and convincing images of the edge magnetic state of a FeGe nanostructure.”

We emphasize that Ref. 4 is only a theoretical prediction. We have cited it as Ref. 17 in the revised manuscript, in order to explain the origin of the chiral edge twist.

Question 3:

Regarding methodology, authors use discrete spin lattice model containing only ferromagnetic exchange, Dzyaloshinskii-Moriya, and Zeeman energy terms. They minimise the system's energy using conjugate gradient method. However, at several places in the main text and supplementary information they refer to "dynamical simulations" and show intermediate states (Supplementary Fig. d and e). It is important to emphasize that the "path" system takes to minimise its energy in conjugate gradient method does not correspond to the real path in the energy landscape if simulated using Landau-Lifshitz-Gilbert dynamics. Therefore, claiming that these states are intermediate states or that dynamic simulations are used is not justified.

Our response:

We agree with this comment and have corrected the corresponding sentences. In the Methods/ Numerical simulations section on pages 17 and 18, we have written:

“Moreover, an NCG method was used to simulate the field-driven magnetization process of skyrmion formation from the helical state. In this case, the initial magnetic state was set to be spin helices and the external magnetic field was then increased gradually so that a metastable state with a local minimum energy was obtained

(Supplementary Fig. 5). This procedure is different from a conventional Landau-Lifshitz-Gilbert (LLG) dynamics simulation⁴⁶, in which the magnetization process is obtained by solving the LLG equation. However, in a certain sense, the NCG method can be referred to as pseudo-dynamic, meaning that at each iteration step it follows the direction of the lowest energy gradient, but without a pre-defined time step.

We have also replaced the words “dynamic simulation” and “intermediate state” by “direct energy minimization” and “incomplete skyrmion state”, respectively, in the main text and supplementary information.

Question 4:

In order to allow reproducibility of simulation results, J and D parameters should be stated. Also, omitting magnetocrystalline anisotropy and dipolar interactions in FeGe needs justification.

Our response:

We thank the referee for this suggestion. In the revised manuscript, we have added the following sentence to the Methods/ Numerical simulations section on page 17

“Taking into account the expression for L_D in (2) in the calculations, we set the coupling constants in arbitrary units of energy to be: $J = 1$, $D = 2\pi/70 \approx 0.09$ and the applied field H in units of saturation field H_D according to (3).”

On page 16, we have written:

“Following a standard approach, we ignore the contribution of magnetocrystalline anisotropy³², which in the case of FeGe is negligibly small with respect to other energy terms, especially at relatively large temperatures as in our experimental setup⁴⁴. Because of mainly Bloch-type modulations of the magnetization in FeGe, the stray field effect can be described exclusively in terms of so-called surface magnetostatic charges⁷, which in the case of a nanostripe produce a closed flux of the stray field mainly outside the sample. Therefore, we expect a relatively small contribution from magnetostatic interactions and in the first approximation we neglect it. Nevertheless, in the most general case and for a precise quantitative description of the system the contribution of stray field interactions should be taken into account.”

We have added Ref. 44 to the revised version of the manuscript, where the effect of the temperature dependence of the weak cubic anisotropy of FeGe is discussed.

In the revised version of the manuscript, we have added numbering to the equations for the equilibrium period of a spin spiral L_D and the saturation field H_D . We have also fixed a misprint in Eq. (3).

Question 5:

Authors state that “transitions between magnetic phases were determined by comparing their energies”, but no explanation on how the system was initialised

(what states were relaxed) is given. At zero temperature, initial state used in the conjugate gradient method is crucial.

Our response:

We thank the referee for this suggestion. In the revised manuscript, we have added the following text to the Methods/ Numerical simulations section (to the last paragraph on page 17):

“As an initial state for the helicoid and conical phases, we used an ordinary Bloch-type spiral with a propagation vector along the long side of the nanostripe (x-axis) or perpendicular to it and along the applied field (z-axis), respectively. For the skyrmion phase, we used as the initial configuration a simple ferromagnetic state with its magnetization along the direction of the applied field (positive direction of the z-axis) and cylindrical domains with opposite magnetization. At each point on the phase diagram of magnetic states shown in Fig. 4, we identified the equilibrium period of the spirals and the equilibrium density of skyrmions in the system, which correspond to the lowest energy density of the particular phase and then compared the energies of all competing phases.”

Question 6:

Authors make several too optimistic and in my opinion not justified claims. In abstract they state that "The flexibility and ease of formation of geometrically confined magnetic skyrmions that we describe has immediate implications for the design of skyrmion-based memory devices". Also, they claim that "the controlled formation of a well-defined number of skyrmions in a nanostripe whose width lies in the range, $L_d < W_y < 2L_d$, can be achieved simply by varying the applied magnetic field". These claims need more detailed explanation, especially how "controlled formation of a well-defined number of skyrmions" can be practically achieved.

Our response:

We have changed the last sentence in the abstract (page 2) as follows: *“The flexibility and ease of formation of geometrically confined magnetic skyrmions that we describe may help to optimize the design of skyrmion-based memory devices.”*

We have also added the following text to the last paragraph (on page 14):

“For instance, by applying a magnetic field locally to a nanostripe with a fixed length equal to an integer number of spin spiral periods $W_x = N \cdot L_D$, one can reliably initiate the nucleation of N skyrmions, which then can be pushed onto a skyrmion track by applying an electric current. Moreover, the range of nanostripe widths $L_D < W_y < 2L_D$ can be considered as a range for the width of a skyrmion racetrack device. Indeed, according to our observations and theoretical modeling, for W_y outside this range we observe an instability of a single skyrmion chain in the form of a collapse of individual skyrmions when $W_y \ll L_D$ or an instability in the form of the formation of a zigzag chain of skyrmions when $W_y \gg 2L_D$.”

Question 7:

In summary, although there are no major flaws that would compromise the reported results, the manuscript needs a more detailed overview of already published literature and details on the simulation method.

Our response:

We appreciate the referee's careful reading of the manuscript and constructive advice. In response, we have made extensive changes to the text, focusing on the viewpoint of already published literature and details of our simulated method.

Response to Reviewer #2

What are the major claims of the paper?

The authors have imaged the magnetic induction from skyrmions in a nanostripe of FeGe, under different conditions, showing that these will take on elliptical form in a predictable manner depending on the applied field and width of the nanostripe. From these images they create a magnetic phase diagram, which is in agreement with a theoretically calculated phase diagram for the experimentally realized conditions. This agreement allows the authors to use their theoretical calculation to predict the states in a much wider range of fields and stripe widths.

Furthermore, the authors show clear and convincing images of the edge magnetic state of the FeGe nanostripe, using the off-axis holography TEM technique, confirming previous suggestions of a twisted edge state. They also show a clear mechanism for the nucleation of skyrmions from helical states in the nanostripe. This control over the formation of skyrmions is vital, if they are to be used for technological applications.

Are they novel and will they be of interest to others in the community and the wider field?

The clear and convincing imaging of ellipticity, skyrmion formation and magnetic edge states are novel and likely to be of great interest to the skyrmion community. The ability to control the formation of skyrmions is vital for proposed technological applications.

Is the work convincing, and if not, what further evidence would be required to strengthen the conclusions?

The authors are to be commended on their excellent description of the experimental method of electron holography and the clear explanation of how they arrived at the images of magnetic induction shown in the paper through subtraction of the electrostatic potential. This detailed description of the experimental methods (appropriately in the supplementary information) help make the data very convincing. The clear magnetic induction maps, and the comparisons with theoretical calculations, make the conclusions very convincing.

On a more subjective note, do you feel that the paper will influence thinking in the field?

I believe the discussion of ellipticity of the skyrmions, and the possibility of these elliptical states having different excitation modes to circular skyrmions in response to alternating fields are likely to give rise to further study. The control of skyrmion formation is a key technological aspect, and other researchers can reproduce it in other samples it is an important step towards applications.

Our response:

We thank the referee for the very encouraging assessment of the novelty and importance of our work.

Please feel free to raise any further questions and concerns about the paper.

Question 1:

There are a number of instances of “Error! Reference source not found” in the paper where references should be; these should be checked and the correct reference number and information inserted.

Our response:

We apologize for this error. In the revised manuscript, we have corrected the references and reference numbers.

Question 2:

The parameter W_y is not explicitly defined when it is first presented on page 5; from the context this seems to be a critical wedge width for circular skyrmion formation, but it would improve clarity to have this explicitly stated at this point.

Our response:

Thank you very much for pointing this out. In order to clarify the corresponding text, we have deleted the original expression and written the following sentence (first paragraph on page 6) in the revised manuscript:

“For nanostripe widths W_y close to a critical value W_y^c , skyrmions exhibit circular shapes ($a = b$), while for $W_y < W_y^c$ and $W_y > W_y^c$ they are predicted to show longitudinal ($a > b$) and transverse ($a < b$) ellipticity, respectively. On further increasing the nanostripe width above a second critical value W_y^t , the skyrmions are expected to arrange themselves in the form of a zigzag chain at equilibrium.”

Question 3:

In figure 3b and d the experimental and simulated values of long and short axes are shown, presumably as the data points. There are also lines in these figures; the authors should explain what these lines are. This is especially important for figure 3b, where the lines do not go through the data points, so presumably represent some sort of fit.

Our response:

The lines are plotted just to guide the eye and no fitting is used. In order to clarify this point, we have written the following sentence to the caption to Fig. 3 on page 20:

“b, Experimentally measured values of a and b determined from phase images of skyrmions and plotted as a function of nanostripe width are shown as symbols, with thin solid lines plotted to guide the eye.”

Question 4:

An explicit reference to figure 4 (the width-field magnetic phase diagram) on line 178 of page 8 should be included; otherwise there is only one parenthetical reference to figure 4a, which is one of the main findings of the paper.

Our response:

We thank the referee for this suggestion and have referred to Fig. 4 at the suggested point in the manuscript. We have also referred to Fig. 4a in the following sentence:

“Based on all of our experimental results (Supplementary Figs 6 and 7), we constructed a width-field (W_y^ -H) magnetic phase diagram for the FeGe nanostripe (Fig. 4a).”*

Question 5:

We would also be grateful if you could comment on the appropriateness and validity of any statistical analysis, as well the ability of a researcher to reproduce the work, given the level of detail provided.

I believe the level of detail provided regarding the electron holograph acquisition and data analysis is sufficient for a researcher with the appropriate equipment and knowledge of the techniques to be able to reproduce the work. The equations and materials parameters used for the theoretical calculations should also provide enough information for someone to reproduce these results.

Our response:

We thank the referee for providing this answer to the question raised by the Journal.

In conclusion, I recommend that this manuscript is accepted for publication, after the minor changes mentioned within this report.

Our response:

We appreciate the referee’s recommendation to publish our manuscript. We have made all of the requested revisions.

Response to Reviewer #3

The paper from Jin and co-workers reports the observation of elliptical distortions of

individual skyrmions in a nonassertive of FeGe. The experiments are carried out via state-of-the art electron holography in a TEM.

On the technical aspects of the manuscript, I have minor objections. I found the data beautiful and their interpretation mostly believable. The authors observe very clearly the deformations of the skyrmions and produce a phase diagram for the nanostripe by investigating a large parameter space of temperatures and magnetic fields.

The results are well described and presented, with high quality and clear images. Scientifically, the observation of these distortions is of great value for the description of topological magnets, and so is the observation of the chiral edge twist reported here for the first time.

Our Response:

We thank the referee for the very positive comments about the value of our work.

I have some criticism on the following aspects of the paper:

Question 1:

the authors put a lot of emphasis on the impact that these results may have for applications in data storage devices. While I agree that this kind of research is intended to seed new ideas in spintronic devices, one should be careful in not overselling rather fundamental research for non-realistic applications. I believe that the authors should make an effort in the introduction to explain a bit better to the general reader what really is the relevance of this contribution for applications. For example, they mention passing by that the race-track memory method could benefit from nanostripes supporting skyrmions. A little explanation on why and how would not harm here.

Our Response:

Please see our response to question 1 by Reviewer #1. As stated above, we have made extensive changes to the text by providing a detailed overview of the published literature on skyrmions in confined geometries in the revised manuscript.

In addition, in order to clarify our statements about the importance of our findings, we have added the following text to the conclusion of the manuscript (last paragraph on page 14):

“Moreover, the range of nanostripe widths $L_D < W_y < 2L_D$ can be considered as a range for the width of skyrmion racetrack device. Indeed, according to our observations and theoretical modeling, for W_y outside this range we observe an instability of a single skyrmion chain in the form of a collapse of individual skyrmions when $W_y \ll L_D$ or instability in the form of the formation of a zigzag chain of skyrmions when $W_y \gg 2L_D$.”

Question 2:

Both in the abstract and the introduction, the phrase "width in the range of only a few

tens of nanometers" is used. In spoken english, the phrase "a few" is conventionally taken to mean "three". The width of the regions analysed in the range 45-150nm, much greater than a few tens

Our response:

The sentence "...a few..." has been modified to "...40 -150 nm...".

Question 3:

At the bottom of page 3 the authors aim to put their electron holography measurements into context by mentioning Fresnel imaging (which has been much used to image skyrmions and references should be given). The neglect to mention that the Lorentz STEM technique of DPC has been applied to discover aspects of skyrmion structure with nanometer spatial resolution. Such as in:

1) McGrouther et al., arXiv:1606.04681v2

2) Matsumoto et al, Sci. Adv. 2016, 2:e1501280

Both articles show the possibility for distortions of skyrmions structure, in particular the latter, which highlights heavily distorted skyrmions at a crystal grain chirality boundary, similar to states shown by the authors.

Our response:

Both papers are now cited in the revised manuscript.

In the first paragraph on page 7, we have written:

“Significantly, the skyrmions can be seen directly to adopt a sequence of compressed, regular and stretched morphologies with increasing nanostripe width, fully consistent with the theoretical prediction shown in Fig. 1a. Similar distorted skyrmions were recently reported in two-dimensional $\text{FeGe}_{1-x}\text{Si}_x$ samples as a result of local lattice disorder around the edge of a crystal grain²⁴.”

24. Matsumoto, T. et al. Direct observation of $\Sigma 7$ domain boundary core structure in magnetic skyrmion lattice. *Sci. Adv.* **2**, e1501280 (2016)

In the last paragraph on page 9, we have written:

“Recent experimental studies using differential phase contrast (DPC) imaging performed in a scanning TEM revealed an intrinsic six-fold symmetry of the internal structure of skyrmion lattice cells in two-dimensional FeGe crystals³⁵. We have also observed this hexagonal skyrmion structure in two-dimensional FeGe by using off-axis EH³⁶. These observations confirm that the shapes of isolated skyrmions can be modified significantly in confined geometries.”

35. McGrouther, D., et al., Internal structure of hexagonal skyrmion lattices in cubic helimagnets. *New J. Phys.* **18**, 095004 (2016)

Question 3:

Later, at the top of page 5, the authors over-generalise their description of chiral

magnetic phenomena. When describing the behavior in bulk crystals they completely neglect to mention the existence of the A-phase pocket. When describing behaviour in thin crystal sections, care is needed to not refer to their sample as a film. Also, their description of the form of the skyrmion in the lattice contradicts that shown by McGrouther et al., but also one of their own works, arXiv:1606.05723v1, which shows that the skyrmions possess hexagonal not circular character. Of course the case might be different for isolated skyrmions and those shown by the authors when they are not organized in a triangular lattice. However, the electron holography images appear far too heavily filtered to be able to conclusively make a confident judgement in this case.

Our response:

We thank the referee for this suggestion. We have included additional content about chiral magnetic phenomena in the revised manuscript. In the first paragraph on page 5, we have written:

“In a bulk crystal, the spin helix generally evolves into a conical phase and then into a field-saturated ferromagnetic state in the presence of an increasing external magnetic field. Skyrmions appear in the form of a lattice and occupy only a tiny pocket in the magnetic field H and temperature T phase space, as the temperature is slightly below the Curie temperature T_c . In a thin crystal of a chiral magnet, the stability of a skyrmion lattice phase will be significantly enhanced^{6,25} due to uniaxial anisotropy²⁸ or spatial confinement^{29,30}. In both bulk compounds and films, skyrmion crystals have a fixed lattice constant and adopt approximately circular shapes.”

With regard to the internal structures of magnetic skyrmions, we have also observed hexagonal skyrmions in two-dimensional FeGe using off-axis EH. This observation implies that the shapes of isolated skyrmions can be significantly modified in confined geometries. We have added this content to the last paragraph on pages 9 and 10. We have also changed the words “*approximate circular shape*” into “*hexagonal shape*”.

Question 4:

Some references have not compiled in the pdf file, just run a check of the bibliography.

Our response:

We apologize and have corrected these errors.

Question 5:

Overall, I find this paper of good quality. The data are obtained with a state of the art TEM for electron holography and they address an issue of current interest in condensed matter physics. I am not that sure about the interest of such a report for an audience that goes beyond the community working on magnetic skyrmions. If the authors can provide a clearer motivation for such a broad interest and other

reviewers, not directly working in the field of magnetic skyrmions imaging, feel that this article deserves the attention of a broader community of physicists I would have no further objections once the above has been taken care of.

Our response:

We appreciate the referee's assessment of the quality of our paper. On the basis of question 1 raised by reviewer 1#, we have made extensive changes to the introduction of the paper to highlight the importance and novelty of the work. The fact that the observation of skyrmion distortions is of great value for the description and understanding of topological magnets is expressed in the first paragraph on page 5. In addition, to highlight the broad interest of the work, we have written in the last paragraph on page 12:

“Our results are based on the application of state-of-the-art electron holography in combination with advanced computer simulations based on atomistic models and direct energy minimization. In this context, our experimental results could not have been obtained using other methods such as small angle neutron scattering, transport measurements³⁹ or Lorentz transmission electron microscopy⁶. From a methodological perspective, the experimental approach that we use offers the prospect of allowing two and three-dimensional magnetic textures in other nanometer-scale spin systems to be quantified⁴⁰ in future studies. Such a capability is currently lacking using any other technique. In a broader context, our experimental results and theoretical analysis therefore open new avenues for exploring quantum confinement in other complex non-linear spin systems.”

Based on the novelty and importance of our results and on the comments of the reviewers, we strongly believe that our work meets the high standard demanded by Nature Communications.

Reviewers' comments:

Reviewer #1 (Remarks to the Author):

I thank the authors for answering the questions I previously raised. However, authors did not resolve some of the main concerns I expected to be addressed:

1. My greatest concern is still about the novelty and significance of the presented work, which is required for the publication in Nature Communications. More precisely, in May last year, authors published a very similar work in the same journal ([18] Du et al. Nature Communications 6, 8504 (2015)) where they studied nanowires with constant width (same material, same/similar technique - Lorentz TEM, very similar system). In that work [18], instead of exploring the wedge-shaped nanowires, they varied the constant width of a nanowire and reported:

- (i) a single skyrmion chain in Figure 1 of Ref. 18,
- (ii) magnetisation tilting at the boundary ("Spins on the two edges tilt onto the plane, and orient their planar components parallel to the boundary" on page 2 of Ref. 18) and clearly visible in Fig. 1 [18],
- (iii) one-to-one transformation of the helical state to the skyrmion chain on page 5 of Ref. 18,
- (iv) transversal ellipticity of skyrmions in nanowires, which is clear from Fig. 1 (c), (j), and (m) of Ref. 18.

These are just some of the findings from the previous work that are reported again in this manuscript for a slightly different system, where a similar behaviour is expected. This is the main reason I asked for a more detailed comparison of the findings in the new manuscript with their already published work, because I do not find the results in this manuscript particularly novel, surprising, or significant. Also in the answer to question 2, authors confirm that this work is not actually the "first observation of the magnetisation tilting at the edge", which further reduces the novelty of the submitted work. Finally, the authors have withdrawn most of the claims about the significance and impact of their work in terms of applications for the development of data storage devices (also raised by Reviewer #3 in Question 1).

2. In the answer to the third question, authors agreed that using conjugate gradient method is not the same as using Landau-Lifshitz-Gilbert dynamics in simulations and that using this method cannot provide intermediate magnetisation configurations in the transition between states. However, in the revised manuscript, authors just changed the terminology and kept the intermediate convergence configurations as true intermediate states. I can maybe agree that the system would probably take "a similar" path in the energy landscape, but because we cannot claim that with certainty (using conjugate gradient method), these configurations are unfortunately meaningless.

Because I still do not find the results in this manuscript novel, surprising, and of great significance, I cannot recommend the acceptance of the manuscript as an article in Nature Communications, unless authors are able to clearly identify and emphasise the novelties when compared to previous works.

Reviewer #2 (Remarks to the Author):

The authors have completed all the minor corrections I suggested in my original referee report, as described in their response letter. The changes/corrections made in response to the other referee reports seem to adhere to the same high standard. Especially the substantially updated introduction should make the paper even more accessible to non-specialist readers.

My original comments about the scientific value and rigour, as well as interest in the topic still stand. Therefore, I am very happy for the paper to be published in this new and improved form.

Reviewer #3 (Remarks to the Author):

I am satisfied by the replies of the authors to my comments as well as to those of other referees. I believe that the paper improved thanks to the constructive criticism raised by all referees and I have no further objection to its publication.

Response to Reviewer #1

Question 1: *...in May last year, authors published a very similar work in the same journal ([18] Du et al. Nature Communications 6, 8504 (2015)) where they studied nanowires with constant width (same material, same/similar technique - Lorentz TEM, very similar system). In that work [18], instead of exploring the wedge-shaped nanowires, they varied the constant width of a nanowire and reported:*

- (i) a single skyrmion chain in Figure 1 of Ref. 18,*
- (ii) magnetisation tilting at the boundary ("Spins on the two edges tilt onto the plane, and orient their planar components parallel to the boundary" on page 2 of Ref. 18) and clearly visible in Fig. 1 [18],*
- (iii) one-to-one transformation of the helical state to the skyrmion chain on page 5 of Ref. 18,*
- (iv) transversal ellipticity of skyrmions in nanowires, which is clear from Fig. 1 (c), (j), and (m) of Ref. 18.*

These are just some of the findings from the previous work that are reported again in this manuscript for a slightly different system, where a similar behaviour is expected. This is the main reason I asked for a more detailed comparison of the findings in the new manuscript with their already published work, because I do not find the results in this manuscript particularly novel, surprising, or significant.

Our Reply: We apologize that we did not respond satisfactorily to the questions raised in the Reviewer's first report and are grateful for the additional questions.

We begin with some general remarks about the novelty of our manuscript with respect to earlier work, in particular Ref. 18, where the observation of a single skyrmion chain and an edge-mediated mechanism of skyrmion nucleation was reported. The smallest value of nanostripe width in Ref. 18 was 130 nm, which is approximately twice the size of skyrmions in FeGe. We believe that most experts working in the field of magnetic skyrmions wish to know how narrow a nanostripe is able to accommodate magnetic skyrmions. In our new manuscript, we show that skyrmions can be accommodated in nanostripes whose width is significantly smaller than the typical skyrmion size by adjusting their shapes *via* elliptical distortions.

We agree with Reviewer#1 that we did not emphasize this statement in our manuscript precisely. In the revised version of the manuscript, after the sentence:

"Therefore, the next important step is to develop approaches that can be used to control the fine structures of individual skyrmions and to nucleate them in a simple manner in a nanostripe, whose size is similar to that of the skyrmions themselves."

We have added the following text between lines 64 and 66:

"The present study is also directly connected to the question: What is the smallest width of nanostripe, which is still able to host magnetic skyrmions over a wide range of applied fields and temperatures?"

We also wish to note that conventional Lorentz transmission electron microscopy, which was used in Ref. 18, *cannot* be used for quantitative magnetic measurements of

objects that have a lateral size of only a few tens of nm. The technique of off-axis electron holography, which is used in the present work, is currently the only method that provides high quality quantitative images of magnetic contrast in nanoscale objects of this size in transmission. In particular:

1. We use state-of-the-art off-axis electron holography to overcome the limitations of Lorentz microscopy for studies of nanoscale chiral magnets. The quantitative measurements of the sizes of skyrmions and their elliptical distortions that we present in our new manuscript could not have been obtained using any other technique, including Lorentz microscopy, which was used in Ref. 18. We have now emphasized these points in the main text between lines 67 and 87.
2. Our experimental approach, which involves the use of a complicated multi-stage sample preparation technique, allows us to construct a phase diagram for a wide range of nanostripe widths based on results obtained from a single sample. Without the use of a carefully prepared wedge-shaped sample, the construction of such a phase diagram would not only be extremely time consuming (as one would have to prepare and make measurements from a large number of different nanostripes, as in Ref. 18), but it would also be less precise because consistent control over the sample thickness and other geometrical parameters (such as the thicknesses of damaged non-magnetic layers resulting from focused ion beam milling) for each sample would be extremely complicated. The wedge-shaped sample in our experiment allows all of these problems to be overcome. The preparation of such a sample, in which one dimension is on the order of μm and the other below 100 nm, is challenging and was not attempted in Ref. 18. We have highlighted these points in the main text between lines 120 and 127.
3. Our experimental phase diagram of magnetic states agrees very well with a theoretical diagram constructed using the most powerful and precise up-to-date computational technique (direct energy minimization on a GPU for models of several million spins; see Figs 3 and 4 in the main text). We therefore not only report on elliptical skyrmion distortions, but also provide comprehensive experimental and theoretical descriptions of their behavior (see Fig. 5 in the main text), which were not previously available. We have highlighted these points in the main text between lines 197 and 271.

In summary, by applying state-of-the-art off-axis electron holography in combination with atomistic spin dynamics calculations, we find that highly geometrically confined skyrmions are able to adopt a wide range of sizes and ellipticities in order to match the geometry of a nanostripe in which they are contained and that they can be created from a helical state with a distorted edge twist in a simple and efficient manner.

The new physics that we describe is novel, significant and dramatically different from that observed in bulk materials, thin films or nanostripes of larger width, as in Ref. 18.

The skyrmions that we observe in our nanostripe of width 45-150 nm are indeed consistent with the previously reported behavior in larger nanostripes of fixed width (> 130 nm) in Ref. 18. However, this is not the central point of our work. In order to avoid any confusion, we have revised the manuscript carefully as follows:

Regarding item (i): *a single skyrmion chain in Figure 1 of Ref. 18*

Our Reply: We discuss the stability of skyrmions in a nanoscale guiding track, whose width is equal to or smaller than the skyrmion size. Figure 1 in Ref. 18 could only have been used to plot one point in the phase diagram that we introduce in Fig. 4.

In order to clarify these points, we have added the following words and sentences to the manuscript:

Line 2: “*geometrically confined magnetic skyrmions*” has been changed to “*highly geometrically confined magnetic skyrmions*”.

Line 52: “*FeGe nanostripes*” has been changed to “*FeGe nanostripes of fixed width*”.

Line 53: The sentence “*However, the widths of the samples in these studies were all larger than the corresponding single skyrmion size.*” has been added.

Line 57: The sentence “*Despite these experimental results, the formation and manipulation of skyrmions has never been studied in detail in confined geometries of sufficiently small dimension.*” has been added.

Regarding item (ii): *magnetisation tilting at the boundary (“Spins on the two edges tilt onto the plane, and orient their planar components parallel to the boundary” on page 2 of Ref. 18) and clearly visible in Fig. 1 [18]*

Our Reply: The edge twist cannot be measured quantitatively using Lorentz microscopy, as in Ref. 18. The edge state observed in Fig. 1 in Ref. 18 is a mixture of a real magnetic signal and an artefact arising from the presence of a thickness variation at the specimen edge. We would also like to emphasize that we do not consider the edge twist to be a central part of our manuscript. We refer to it because it is essential for understanding the mechanism of the longitudinal elliptical distortion of skyrmions, as shown in Fig. 5 in the main text of our manuscript.

Regarding item (iii): *one-to-one transformation of the helical state to the skyrmion chain on page 5 of Ref. 18*

Our Reply: A one-to-one transformation of a helical state to a skyrmion chain is indeed observed in both studies. Our results confirm that the formation of skyrmions in a *highly confined* geometry (in a nanostripe with a width of 45-150 nm) is consistent with the previously reported behavior in Ref. 18. (in a nanostripe with a width of >130 nm). For clarity, we have added the following sentence:

Line 173: “*This special helix-to-skyrmion transformation has been observed in nanostripes with widths above 130 nm¹⁸. Here, we confirm that the same behaviour is followed in a much narrower (45-150 nm) nanostripe.*”

Moreover, the possible applications of this transformation were not discussed in Ref. 18.

Regarding item (iv): *transversal ellipticity of skyrmions in nanowires, which is clear from Fig. 1 (c), (j), and (m) of Ref. 18.*

Our Reply: This consistency is expected because the states illustrated in Fig. 1 in Ref. 18 are within the range of parameters of the phase diagram in the present manuscript. Here, we provide experimental and theoretical estimates of the critical field and the full range of widths of a nanostripe in which this effect can be observed. In order to avoid confusion, we have added the following sentence:

Line 105: *“Recent Lorentz TEM observations of an FeGe nanostripe with a fixed width of 130 nm suggest that elliptical skyrmions can be supported in such a confined geometry.”¹⁸*

Question 2: *In the answer to the third question, authors agreed that using conjugate gradient method is not the same as using Landau-Lifshitz-Gilbert dynamics in simulations and that using this method cannot provide intermediate magnetisation configurations in the transition between states. However, in the revised manuscript, authors just changed the terminology and kept the intermediate convergence configurations as true intermediate states. I can maybe agree that the system would probably take "a similar" path in the energy landscape, but because we cannot claim that with certainty (using conjugate gradient method), these configurations are unfortunately meaningless.*

Our Reply: We understand from this remark from Reviewer#1 that in the previous version of the manuscript we did not fully comprehend his/her previous remark 3 and that the corresponding suggestions and our reply left the reviewer unsatisfied.

We discuss here essentially Fig. 5d in the Supplementary Information. It is well known that topological charge is a constant of motion in the space of continuous functions. Therefore, by changing the spin texture from a ferromagnetic or collinear state to a skyrmion state, we change this topological charge. This is of course an exciting moment, which can be realized in nature by defects or edges initiating Bloch points or other discontinuous states, facilitating the switch of the topological charge.

While we cannot directly capture this moment of transition numerically because of the different possible sources facilitating the transition, it was our motivation to provide the reader with an intuitive explanation of how such a transition can occur, for example at the edge.

Other arguments that motivated us to show Fig. 5d in the Supplementary Information were:

(i) The fact that the path that the system takes in our calculations is consistent, in general, with that observed experimentally.

(ii) To provide comprehensive data to make our results reproducible and verifiable, meaning that any reader who wishes to reproduce our results applying the same calculation method can use our extended data as a reference.

It is not our aim to mislead the reader or to state that the intermediate state presented reproduces fully the intermediate state in the experiments, which most probably appears due to the presence of defects.

Therefore, in the present version of the manuscript we have stated in very clear words:

Line 232 in the main text: “*In Supplementary Fig. 5d, we show qualitatively similar incomplete skyrmions obtained during energy minimization using the NCG method. However, it has to be mentioned that, strictly speaking, such magnetization states obtained using a direct minimization method are not a true physical realization and only final magnetization configurations corresponding to the equilibrium state can be compared to the experimental data.*”

We fully agree with the statements of the Reviewer regarding *Landau-Lifshitz-Gilbert (LLG) dynamics*.

Nevertheless, we also wish to note that the LLG equation is also only a mathematical model and it is not appropriate to unequivocally consider results obtained using the LLG equation as being physically absolutely or more relevant than ours. Indeed, the exposure time in off-axis electron holography is on the order of seconds, while typical timescales that are achievable in simulations based on the LLG equation for such geometrical scales are on the order of nanoseconds. It is clear that the conjugate gradient method has its own advantages and disadvantages. However, the discussion of this subject is beyond the scope of the present work. It was chosen because up to now it is *de facto* one of the most efficient methods for the analysis of equilibrium states.

We sincerely hope that Reviewer #1 agrees with this statement and our corrections, which we have made according to his/her suggestions.

Note: In order to ease reading of the revised manuscript, we have introduced our changes directly into the PDF version.

REVIEWERS' COMMENTS:

Reviewer #1 (Remarks to the Author):

I thank the authors for addressing my concerns from the previous review. In the revised manuscript authors gave more credit to their previous work (Du et al. Nature Communications 6, 8504 (2015)) and made much more clear what are the similarities with their previous work (which was hardly cited in their initial manuscript).

In terms of the micromagnetic simulations, in the rebuttal letter, authors mention that: "It is well known that topological charge is a constant of motion in the space of continuous functions. Therefore, by changing the spin texture from a ferromagnetic or collinear state to a skyrmion state, we change this topological charge. This is of course an exciting moment, which can be realized in nature by defects or edges initiating Bloch points or other discontinuous states, facilitating the switch of the topological charge". This claim is true only when skyrmion states are topologically protected. More precisely, in geometrically confined systems (such as nanowires), skyrmion configurations are not topologically protected and a transition between uniform and skyrmion state would not be facilitated via singularity (Bloch Point) - they can always be created or destructed at the boundary. For instance, see [Cortés-Ortuño et al. arXiv, 1611.07079 (2016)]. Also I cannot agree with the author's claim in the rebuttal letter that they are using "the most powerful and precise up-to-date computational technique (direct energy minimization on a GPU for models of several million spins)" because "powerful" and "precise" are very relative terms in computational science. However, these claims are not made in the manuscript where authors emphasised: "...it has to be mentioned that, strictly speaking, such magnetization states obtained using a direct minimization method are not a true physical realization and only final magnetization configurations corresponding to the equilibrium state can be compared to the experimental data" and I agree with that.

Although I still do not see a novelty which is significant or surprising enough for the publication in Nature Communications, I assume this paper will still attract some interest in the community - as noted by the other two reviewers. Also, in the revised manuscript authors made clear how intermediate states obtained using conjugate gradient method can be understood. Therefore, I do not have any further objections to raise and I can recommend the publication of this work.

Response to Reviewer 1#

Comments:

I thank the authors for addressing my concerns from the previous review. In the revised manuscript authors gave more credit to their previous work (Du et al. Nature Communications 6, 8504 (2015)) and made much more clear what are the similarities with their previous work (which was hardly cited in their initial manuscript).

In terms of the micromagnetic simulations, in the rebuttal letter, authors mention that: "It is well known that topological charge is a constant of motion in the space of continuous functions. Therefore, by changing the spin texture from a ferromagnetic or collinear state to a skyrmion state, we change this topological charge. This is of course an exciting moment, which can be realized in nature by defects or edges initiating Bloch points or other discontinuous states, facilitating the switch of the topological charge". This claim is true only when skyrmion states are topologically protected. More precisely, in geometrically confined systems (such as nanowires), skyrmion configurations are not topologically protected and a transition between uniform and skyrmion state would not be facilitated via singularity (Bloch Point) - they can always be created or destructed at the boundary. For instance, see [Cortés-Ortuño et al. arXiv, 1611.07079 (2016)].

Also I cannot agree with the author's claim in the rebuttal letter that they are using "the most powerful and precise up-to-date computational technique (direct energy minimization on a GPU for models of several million spins)" because "powerful" and "precise" are very relative terms in computational science. However, these claims are not made in the manuscript where authors emphasised: "...it has to be mentioned that, strictly speaking, such magnetization states obtained using a direct minimization method are not a true physical realization and only final magnetization configurations corresponding to the equilibrium state can be compared to the experimental data" and I agree with that.

Although I still do not see a novelty which is significant or surprising enough for the publication in Nature Communications, I assume this paper will still attract some interest in the community - as noted by the other two reviewers. Also, in the revised manuscript authors made clear how intermediate states obtained using conjugate gradient method can be understood. Therefore, I do not have any further objections to raise and I can recommend the publication of this work.

Our response:

We greatly appreciate the referee's assessment of our manuscript and recommendation for publication. We also thank the referee for the information about skyrmion formation in geometrically confined systems, in which singularity is not necessary due to edge effects. Although we did not refer to "the most powerful and precise up-to-date computational technique (direct energy minimization on a GPU for models of several million spins)" in the main text, we believe that this is very powerful micro-magnetic simulated method, especially for such modulated spin textures.